# Predictive Models for Forecasting Public Health Scenarios: Practical Experiences Applied during the First Wave of the COVID-19 Pandemic

**DOI:** 10.3390/ijerph19095546

**Published:** 2022-05-03

**Authors:** Jose M. Martin-Moreno, Antoni Alegre-Martinez, Victor Martin-Gorgojo, Jose Luis Alfonso-Sanchez, Ferran Torres, Vicente Pallares-Carratala

**Affiliations:** 1Department of Preventive Medicine and Public Health, Universitat de Valencia, 46010 Valencia, Spain; jose.l.alfonso@uv.es; 2Biomedical Research Institute INCLIVA, Clinic University Hospital, 46010 Valencia, Spain; martin_vicgor@gva.es; 3Biomedical Sciences Department, Faculty of Health Sciences, Cardenal Herrera CEU University, 46115 Valencia, Spain; antoni.alegre@uchceu.es; 4Orthopedic Surgery and Traumatology Department, Clinic University Hospital, 46010 Valencia, Spain; 5Preventive Medicine Service, General Hospital, 46014 Valencia, Spain; 6Biostatistics Unit, Medical School, Universitat Autonoma de Barcelona, 08193 Barcelona, Spain; ferran.torres@uab.cat; 7Health Surveillance Unit, Castellon Mutual Insurance Union, 12004 Castellon, Spain; pallarev@uji.es; 8Department of Medicine, Jaume I University, 12071 Castellon, Spain

**Keywords:** COVID-19, health policy, public health, explanatory models, forecasting, predictive models

## Abstract

**Background:** Forecasting the behavior of epidemic outbreaks is vital in public health. This makes it possible to anticipate the planning and organization of the health system, as well as possible restrictive or preventive measures. During the COVID-19 pandemic, this need for prediction has been crucial. This paper attempts to characterize the alternative models that were applied in the first wave of this pandemic context, trying to shed light that could help to understand them for future practical applications. **Methods:** A systematic literature search was performed in standardized bibliographic repertoires, using keywords and Boolean operators to refine the findings, and selecting articles according to the main PRISMA 2020 statement recommendations. **Results:** After identifying models used throughout the first wave of this pandemic (between March and June 2020), we begin by examining standard data-driven epidemiological models, including studies applying models such as SIR (Susceptible-Infected-Recovered), SQUIDER, SEIR, time-dependent SIR, and other alternatives. For data-driven methods, we identify experiences using autoregressive integrated moving average (ARIMA), evolutionary genetic programming machine learning, short-term memory (LSTM), and global epidemic and mobility models. **Conclusions:** The COVID-19 pandemic has led to intensive and evolving use of alternative infectious disease prediction models. At this point it is not easy to decide which prediction method is the best in a generic way. Moreover, although models such as the LSTM emerge as remarkably versatile and useful, the practical applicability of the alternatives depends on the specific context of the underlying variable and on the information of the target to be prioritized. In addition, the robustness of the assessment is conditioned by heterogeneity in the quality of information sources and differences in the characteristics of disease control interventions. Further comprehensive comparison of the performance of models in comparable situations, assessing their predictive validity, is needed. This will help determine the most reliable and practical methods for application in future outbreaks and eventual pandemics.

## 1. Introduction

The onset and subsequent spread of an emerging or re-emerging infectious disease have always been a cause for concern. For more than a century, modeling has been used to characterize the evolution, assess the impact of public health interventions, and suggest the optimum course of action to control emerging infectious diseases [1]. Theoretical approaches to investigate and understand what happens around epidemics are generally based on mathematical models [2]. Modeling has made it possible in many cases to predict the behavior of epidemics, which has led to informing key improvements in critical areas such as the organization of national health systems, planning medical supply needs, predicting critical points of overload in health systems, and determining when and to what extent some necessary interventions and countermeasures should be implemented, relaxed or terminated [3].

The study of the transmission, prevention and control of severe acute respiratory syndrome coronavirus-2 (SARS-CoV-2) has concentrated most of the current scientific activity since the first cases of pneumonia were described in Wuhan (People’s Republic of China) at the end of December 2019 [4]. SARS-CoV-2 spread rapidly across all continents, causing an unprecedented global public health crisis that we are still facing [5]. The international response, initially slow because it was assumed to be distant, became astonishingly forceful, with massive restrictions on travel to China and strict controls on citizens returning to China for two weeks, even in the absence of symptoms [6]. Reducing the transmission of severe acute respiratory syndrome coronavirus 2 (SARS-CoV-2) became a global priority then and now [7]. Very soon, the data available from the first affected countries, such as China, Italy or France, were modeled to estimate the timing and magnitude of the epidemic peak of the first wave. COVID-19 behaved like a tsunami, so it has been essential both to understand how it happened (explanatory models) and, based on what happened, to be able to predict the evolution to try to be better prepared for events (predictive models). In a nutshell, these two approaches differ in that while explanatory models seek to identify the risk (or protective) factors that are etiologically related to an outcome, predictive models try to find an optimal combination of factors that best predicts an imminent development or future evolution of the problem [8]. The first models used in the early 2020s emphasized the capacity for anticipation to inform policy decisions, in what is usually called “predictive understanding,” based on the value that scientists place on scientific theorizing to infer conclusions and from this to formulate patterns (developing models) to deduce what needs to be done. However, this proved insufficient from a scientific point of view, as there was still an unmet need to develop explanatory knowledge to change the model by adapting it to emerging evidence [9].

Predictive models were used from the very beginning in COVID-19, trying to anticipate scenarios and forecast epidemic peaks. The predictions even went so far as to estimate very specific details, such as the number of ventilation units that would be necessary for future epidemic peaks as in the study by Fanelli and Piazza in Italy [10]. In this regard, let us point out that health systems should use optimal prediction models based on a sound review of the literature to predict the number of cases and thus support preventive measures, including interventions that may generate some controversy, such as social distancing or, even more so, lockdown or confinement [11]. However, these predictive studies have some limitations, as in the case of the use of an underlying ecological design, and some potentially associated errors, such as non-differential classification and incorrect spatial or temporal assignments [12]. In addition, given the changing viral and immunological dynamics, many predictive models cannot truly estimate the rate of replication and how it affects the population in the coming weeks [13]. Also, since their goal is to make predictions in anticipation of behavioral changes in epidemics, external validation is normally not feasible at the time that it is needed. As a consequence of this, there is a non-negligible risk of overfitting and pseudo-accuracy which could result in high precision for observed data but lower predictive accuracy for new observations [8], which are actually the target of these models.

The models described in the following sections should initially be considered deterministic in nature, by default. However, extensions to stochastic models can be made to include a random component. Stochastic models allow for the estimation of uncertainty (i.e., standard errors) when estimating parameters. The use of this “random framework” is more in line with reality and, therefore, stochastic models are considered more realistic than deterministic models, the latter being only valid for sufficiently large populations, which may be the explanation for possible discrepancies between a deterministic model (even if well-chosen at the outset) and real-life results [14,15].

Modeling made it possible to provide health managers with the available tools to foresee the behavior of the spread of this infectious disease, and thus be able to establish control strategies through simulation. The development of these tools is a multidisciplinary task in which mathematical algorithms are the basis and play a fundamental role. The models of the first wave of COVID-19 (March–June 2020) were of great interest due to the epidemiologic characteristics. This work aimed to review and characterize the modeling approaches used during the first wave of COVID-19 and tried to shed light that could help to understand them for future practical applications.

## 2. Materials and Methods

A comprehensive bibliographic search strategy was performed using the keywords [COVID-19] AND [predictive OR forecasting OR explanatory] AND [models OR modeling] in search sources PubMed, Google Scholar, Cochrane Library and Web of Science; in addition, a search was made in medRvix, BiorXiv and arXiv (Appendix A). Databases were searched for articles published up to 1 April 2022, but whose applied modeling was conducted during the first wave of the COVID-19 pandemic (referring to the period from 1 March to 30 June 2020). The search strategy is fully detailed described in the Appendix A. Inclusion criteria considered studies that apply at least one model in public health, studies that produce a solution to COVID-19, studies that explicitly address the issue of COVID-19, and studies written in English. Exclusion criteria included studies published before 2020, works exclusively published as poster papers and/or extended abstracts, and studies that were not part of the COVID-19 outbreak, studies that mentioned COVID-19 techniques but did not use model and theoretical works without application. The articles retrieved from that search were selected first through a first reading of the title and abstract, and in a second step through the screening of the full text as per the main PRISMA 2020 statement recommendations [16], systematizing the steps in identification, screening, eligibility, and inclusion, all to help structure our work and as a useful basis for the potential reader. We did not complete the PRISMA 2020 model exhaustively, among other reasons because it was not feasible to address some of the 27 questions contained in the checklist, but we did follow the main systematic steps, which helped us to identify the maximum number of published articles and to analyze the observation period considered.

## 3. Results

The flow chart reflecting the literature search and study selection is illustrated in Figure 1.

Once the studies were identified based on the modeling methods used for each one, we distinguished the mathematical models by classifying them into two groups: standard epidemiological models and data-driven models.

### 3.1. Standard Epidemiological Models

Epidemiological models divide the population into several compartments, and differential equations predict movements from one compartment to another. Their main advantages are that they consider the dynamics of the contagion of an infectious disease in the population, they allow modeling numerous variables that may affect spread (quarantines, vaccination, reinfection, isolation) and have high power to predict the worst scenarios. However, they have some weaknesses such as their excessive dependence on parameter estimation, and that these parameters must also be adapted and updated during the epidemic [17]. Some alternative models according to modeled parameters are shown in Figure 2, and detailed explanations of the characteristics and uses of the models are shown in the Table 1.

#### 3.1.1. SIR (Susceptible-Infected-Recovered) Model

This model, originally developed by Ross and Hamer in 1915 [26] and further improved by Kermack and McKendrick in 1927 [27], consists of three coupled non-linear differential equations that allow for the prediction of the transmission of an infectious disease, helping to make decisions and public health interventions. SIR models allow for a quantitative analysis of a policy that is expected to be optimal when applied to different groups. For instance, the model may be helpful to predict differential benefits such as a reduction in economic impact or excess of deaths when applied to different expected risk or age groups. This may allow for the making of different tailored interventions such as a more or less strict lockdown in the oldest rather than in the middle- or younger-aged groups, which might be more optimal than uniform interventions applied to all population groups. Acemoglu et al. praised the SIR model, calling it a “workhorse tool” in the COVID-19 pandemic thanks to its capacity to bring together economic effects and trade-offs depending on the differential risks in the population [28].

The results from the SIR model, when applied without special adjustments to the COVID-19 pandemic, have restrictions due to several reasons. Basically, the indefiniteness of the parameter N (community’s population) was not only conditioned by the behavior of the community that can produce additional waves, but also by the intrinsic limitation of the model, that to have an adequate performance has to be preferably limited to small populations where the results can be properly adjusted. Despite these drawbacks, this model has some advantages, such as its direct and transparent approach, easily implemented and understandable through compartmental relationships [29]. Fanelli and Piazza used this model to successfully forecast the first wave peak in Italy during the first wave, predicting that in future waves the reduction of mortality rate and the slow-down of the epidemic peak will only be visible if restrictive measures are implemented in the first days to make possible an 80–90% reduction in the infection rate [10].

Ahmetolan et al. estimated the basic reproduction number, mean duration of the infectious period and the estimate of the time of the peak of the epidemic wave using an SIR model and data from the early phase of daily detected cases and daily mortality of China, South Korea, France, Germany, Italy, Spain, Turkey, Iran, the United Kingdom and the United States. SIR models were analyzed for each country to fit the cumulative data of infectious cases with an error of 5%. It was observed that the basic reproduction number and the mean duration of the infectious period could only be estimated in cases where the spread of the epidemic was over (for China and South Korea in the present case). Moreover, it was also shown that all peaks and inflection time-points could be robustly estimated from the normalized data. Validation conducted by comparing predictions to actual data showed that as long as lockdown measures were maintained, the predictions held true for all countries except the US [18].

Although this model is widely recognized and it is considered potentially validated, there have been several attempts to improve it with additional epidemiological models after the appearance of COVID-19. Models with additional parameters are shown in the Table 1 and are summarized in a concise manner below.

Cooper et al. added two improvements to the SIR model. First, the total population do not necessarily remain constant. And, second, susceptible individuals do not decline monotonically because there was clear evidence that they could even increase, since the observation of the data on the evolution of COVID-19 in the databases allows us to observe that an increase in the number of infected people (I) results in a surge in the susceptible population candidate for infection (S). They analyzed the spread of the disease in different countries, and the predictions fitted nicely with the published case data in Italy and Texas, but not in China or South Korea, where the number of cases fell very quickly due to stricter preventive measures. Notably, if we compare the published data with the SIR forecast, we can reasonably predict the success of government interventions [30].

#### 3.1.2. SQUIDER Model

Khan et al. extended the SIR to assess several responses to COVID-19 in eight US states, fitting the reported incidence data jointly with the suppression of prevention measures. They made a distinction between the reported cases and the asymptomatic/mild cases not detected. Also, they included the effect of quarantine, isolation measures and social distancing, as well as the reintroduction of recovered individuals with loss of immunity to the susceptible population. They called this model “SQUIDER”, using an acronym that arises from the seven parameters recognized in this type of modeling, including susceptible individuals (S), social distancing (Q), undetected infected (U), detected infected (I), detected recovered (R), plus undetected recovery/death (E), and detected death (D). This approach has particularly been used to predict future COVID-19 deaths and future COVID-19 cases in several U.S. states based on data from Johns Hopkins University. These authors state that this model describes fairly well the epidemiological data in the US states under scrutiny, pointing out the greater number of compartments with respect to the SIR model and the non-linearity in the infectious power of the disease as the key to success [19].

#### 3.1.3. SEIR Model

SEIR models are widely used to predict possible contagion scenarios by describing infectious disease dynamics in the event of an outbreak, and they are useful in predicting whether preventative measures (such as lockdowns) may be effective. For its evaluation, the total number of patients, the number of patients recovered after the disease and the number of deceased are used. Notably, Hauser et al. showed that the case fatality rate is not a good predictor of the overall SARS-CoV-2 mortality and that it should not be used for policy evaluation or comparison between settings [31]. In the unforced SEIR models the evolution of the contagion does not take into account temporary or seasonal effects that could alter the spread of the disease (such as the school year, vacation periods, mass celebrations on certain dates or other contingencies). In those models, the conditions of the initial infection are more critical in the transmission of the epidemic outbreak than the reproduction number R0. This method allows a better understanding of the speed of transmission of infectious diseases, especially those that are transmitted by water and by vectors [32].

Struben applied a SEIR model that collected not only the dynamics of virus transmission but also political decisions, interventions in response to the epidemic and the social contacts of the populations. This model was able to differentiate mild, asymptomatic and severe cases, in addition to reliably representing the heterogeneity of the different social and demographic segments of the population. The author concludes that it is a solid model for COVID-19, with the advantage of being flexible for other contexts in other infectious diseases [20].

Kuniya proposes the SEIR compartmental model to predict the epidemic peak of COVID-19 in Japan using real-time data from 15 January (first reported case) to 29 February 2020, considering the uncertainty due to the incomplete identification of the infected. The author estimated the basic reproduction number, R0 = 2.6 (95% CI, 2.4–2.8), using a Poisson-noised least-squares method and predicted that the epidemic peak could arrive in early to mid-summer 2020 [33]. The latter prediction was consistent with the WHO statement of 6 March 2020 that it was a false hope that COVID-19 would disappear in the summer like the flu [34].

#### 3.1.4. Time-Dependent SIR Model

Proposed by Chen et al., this model was apparently able to predict the evolution of the pandemic in China with less than 3% of errors. This model tracked the transmission and recovery rate at a given time covering two types of infected persons: detectable and undetectable. This time-dependent model is not as static as traditional SIR models, as it is more dynamic and more robust, able to track the characteristics of recovery rate and transmission rate with respect to time. Finally, they concluded that some measures like social distancing reduce the effective reproduction number [35].

#### 3.1.5. Other Proposed Models

The SEIRS model is a less complex approach, but considers that recovered patients may become susceptible again (susceptible-exposed-infectious-recovered-susceptible). This model provides a potentially good fit to weekly incidence and reproduction figures, and also forecasts that, similar to other coronaviruses, outbreaks will persist each winter for several years with peaks in the second week of January [21]. Bjørnstad et al. have recently developed a web and R program to allow a fast and intuitive application of this model to different pandemic diseases and situations [36].

Calafiore et al. included in the SIRD (Susceptible-Infectious-Recovered-Deceased) analysis parameters such as the initial number of susceptible individuals and the proportionality α factor (i.e., the number of detected positives versus the unknown number of infected individuals) to predict the spread of COVID-19 in Italy. They concluded that it was not was not possible to accurately calculate the variability of the results because of time restrictions, but it was estimated at ±78% based on previous sources [23].

Venkatasen et al. proposed a SIR model for India with herd immunity and a flattening curve based on a predisposed and increasing population and difference in birth and mortality rates, and included key variables such as the number of immune individuals and the number of sensitive individuals. This simulation showed the evolution of susceptible, infected and recovered persons over time, with the transmission rate and fatality rate. However, they stated that the model had some relevant limitations such as the lack of precision of the results and the high dependence on constantly published data [37].

The Weibull distribution model was used to estimate the incidence in the Hokkaido prefecture in Japan [38]. The results yielded figures higher than those officially reported, pointing out the gap between estimated and detected cases to stop undetected transmissions. Stochastic simulations led to the conclusion that the local risk of an outbreak depended on several parameters: the evolution of the number of cases in the country of origin (in this case China), the frequency of travels with other countries and the effectiveness of the measures in the destination country. Travel restrictions were useful in countries with few flights to China and high R0 numbers, while countries with many connections to China and low R0 numbers benefited most from policies aimed at lowering R0 [39].

Rocchi et al. [24] proposed a SIRS (Susceptible, Infected, Recovered, Susceptible) model to forecast the evolution of the epidemic in the province of Pesaro-Urbino, one of the main areas of focus of the epidemic in Italy, under the hypothesis of non-permanent immunity. This approach offers an analytical solution to the problem of finding possible stationary states. However, it is crucial to take into account that the results on recoveries also include the large proportion of asymptomatic subjects.

Other models such as the SUQC (susceptible-unquarantined-quarantined-confirmed) or SIRV (susceptible, infectious, recovered vaccinated) are not considered in this review of alternative models, as the quarantine and post-vaccination periods had not yet been completed at the time of writing this article [40].

### 3.2. Data-Driven Models

Forecasting relies on the use of past data to project future outcomes, and this can be complex in the context of a new or unfamiliar situation, such as the case of the COVID-19 pandemic.

In these models, instead of establishing several compartments and predicting movements by means of differential equations, a predictive curve is used to evaluate validation or readiness concepts, obtaining a good fit to retrospective data, and allowing short-term projections. The collection of several sources of information is carried out to analyze and subsequently carry out a plan and public health policy decisions which imply an evaluation of future scenarios. Consequently, the consideration of the time-span horizon is an unavoidable factor to be accounted for in decision making. Notably, these models are not unique or permanent, so exquisite rigor and permanent review of the predictions are essential. Despite these caveats, it is feasible to use those tools to evaluate, to validate or to predict and plan in a short-term period. These data-driven models are useful, for example, to highlight the potential need of lockdown and self-isolation, as we can see in this early study in Italy in the beginning of the epidemic [41]. However, they do not take into account the dynamics of the spread of the disease, and that is why these models present more limitations when it comes to applying conclusions for long-term policies [17].

#### 3.2.1. Autoregressive-Integrated Moving Average (ARIMA) Models

ARIMA models consist of statistical techniques that allow for the building of a model for a numerical series for which repetitive patterns are found and which does not have excessive random data. They are mainly used in some fields such as in economics, and it is necessary to collect the data at regular and constant intervals [11]. According to Shankar et al., there was no particular model that was substantially superior to others [17].

The first explanatory models began to appear after the first months of the epidemic. Thurner’s team was struck by the fact that infection curves in numerous countries (United Kingdom, Sweden, Finland, Poland, Indonesia, and the United States) had linear growth that could span very long periods, something inexplicable with traditional models that usually start with an S shape. They tried to explain it by attributing this to the fact that these models did not take into account the network-shaped contacts of the population. By imputing about five contacts per person in situations without confinement, and 2.5 contacts per person in confinement situations, they developed an explanatory model that fit with great precision both in countries with early strict confinement (Austria) and in countries without early strict confinement (the United States). Policies on confinement and contact networks limited the spread of infections and must be taken into account, especially in countries whose family structure networks are made up of many individuals [42].

Other studies such as those by Sorci et al. [43] tried to explain the enormous variation that could be found in the case-fatality rate (CFR) from one country to another, with a minimum of 0% and a maximum of 20%. These disparities can be explained by the different state of epidemic control, given that some countries had an earlier onset than others, by the comorbidity of other diseases or by the overload of the hospital system, especially due to the number of intensive care beds. In addition, there is a known bias due to differential procedures with regard to the counting of the number of cases, depending on the capacity of the system to detect cases in the population through mass screening. For example, the number of tests performed per 1000 inhabitants varied from 0.9 (Indonesia) to 179 (Iceland). The CFR was statistically significantly higher in countries with higher levels of disability adjusted life years (DALYs) due to a higher burden of diseases (chronic respiratory and cardiovascular diseases, kidney diseases, cancer), exposure to air pollution and tobacco, age over 70 years, gross domestic product (GDP) per capita or high level of democracy. Conversely, there was an inverse negative association with comorbidity with other lower respiratory tract infections and the number of beds per thousand inhabitants [43].

#### 3.2.2. Machine learning (ML)

This approach consists of computer algorithms that use past experience and data to learn and improve to classify, interpret and understand the data. ML is an emerging technology that can be used for classification, diagnosis, prognosis, regression and even chatbots [11].

Bottino’s team conducted a systematic review to verify the use of artificial intelligence techniques, such as machine learning (ML) and deep learning (DL). Many factors were present in almost all studies, such as age, PCR, and LDH levels. However, they found that numerous values of variables that could be helpful in the prediction of the ML and DL models (vital signs, comorbidities, laboratory results, radiographs) were omitted. Another limitation is that most studies had a marked imbalance of survivors and non-survivors [44]. In this line, Yan et al. found a high prediction of mortality three days after admission [45], which highlights the importance of a rigorous learning in the identification of those patients who present a high risk of mortality. Regarding imaging techniques, they have been used to predict mortality but not in a DL context. To be able to use optimal ML, an improvement in data collection through a good systematic methodology would be necessary [44].

#### 3.2.3. Genetic Evolutionary Programming (GEP) Model

Genetic programming is a variant of a genetic algorithm in which a computer creates a hierarchical tree-like structure to find a relationship between input variables and output variables, based on Karva language. This approach has proven to be more efficient than classical techniques and more stable than artificial neural networks, in addition to generating simple prediction equations that can be optimized over time. This model has been used to predict cases in India by including two main parameters. The first was the confirmed cases and the second the number of deaths. The models were shown to be reliable, satisfied the external validation requirements, and can be further improved by new algorithms, for instance krill herd and naked mole-rat algorithms [13].

#### 3.2.4. Long Short-Term Memory (LSTM) Model

It is a recurring neural network whose structure is in the form of a chain and that instead of having a single neural network layer has four layers, each one performing its own special network function. It is very useful for predicting the number of new cases over a given period and making a realistic forecast over time [11]. This model has also been used in Canada, where it successfully predicted in March 2020 that the first wave would end in June 2020. However, it was not accurate in predicting that the pandemic would end in December 2020 and that it would not last as long as the Spanish flu of 1918 [46].

#### 3.2.5. Global Epidemic and Mobility (GLEM) Model

The GLEM model tries to identify disease compartments and establishes a scenario for simulation using data such as compartment characteristics, transition values, environmental characteristics, etc. This model is being used in studies related to COVID-19 [11]. An example of software based on this model is GLEaMviz, capable of simulating realistic epidemic scenarios that are useful when establishing policies and analyzing the different containment measures [47].

To provide a summary overview of the objectives and conclusions of the relevant epidemiological models applied during the COVID-19 pandemic, Table 1 summarizes the main articles identified and discussed.

## 4. Discussion

We have conducted a literature review which, although not fully comprehensive, provides a sound basis from which we have described and discussed the evidence for different predictive models in the current COVID-19 pandemic. And despite the fact that the review procedures were rigorously performed dually and independently, in this type of review it is usually understandable that a certain risk of bias cannot be completely ruled out. Nevertheless, we believe that none of the possible methodological potential flaws and limitations would have changed the conclusions of the review. The main limitations of this study are largely conditioned by the limitations of the studies reviewed, one of the main ones being the dependence on the good collection and classification of the cases and the reliability of the definition of a positive case, which is not always homogeneous and can lead to biases depending on how rigorous the national health agency is in classifying patients. Another potential methodological limitation has to do with the fact that we used preprint articles in a first phase, in which the peer review process had not initially been completed, although this limitation has been addressed because in the final drafting of this manuscript we were able to ascertain all articles that were subsequently reviewed and published in peer-reviewed journals.

We focused on studies available in the most relevant bibliographic databases. Therefore, government documents and other potentially relevant sources of gray literature are not included in this review. Although we consider that it should not have a relevant impact on our work, we need to make it clear that we may have not included all the information. We have just reviewed the publicly available relevant information to address the fundamental objective of this review.

Future studies, in addition to analyzing the behavior of the different epidemic waves, will have to face other critical effects on health and the economy as a result, for example, of quarantines. They will also have to look at impacts and predictions on quality of life, increased demand for medical care, percentage of chronic patients not cared for, number of people unnecessarily quarantined, unemployment, increase in domestic or social violence, and travel restrictions. These are just some of the variables to be considered in the near future [48].

We have been able to verify how most models on COVID-19 are quantitative techniques which rely on different assumptions about relevant model parameters. They consistently showed that quarantine was important in reducing incidence and mortality during the COVID-19 pandemic, although the magnitude of the effect was uncertain. That said, early implementation of quarantine and the combination of quarantine with other public health measures were seen to be key elements in ensuring the effectiveness of such decisions. Policy makers and public health managers need to constantly monitor outbreaks and the effects of different interventions to obtain an optimal balance in the application of such measures.

It is not easy and at least it would be highly debatable to judge which model might be better. Kırbaş et al. compared LSTM, ARIMA and NARNN and concluded that the LSTM model was more successful than the other two [49]. However, it would be too premature to make definitive conclusions, since there were differences in the available data, in the population behaviors and in the implementation of restrictive measures, which clearly varied among countries and time-periods, therefore further studies are still needed to make conclusions in this sense. Furthermore, different policies can influence the spread of the pandemic and may have a major impact on model results. The same model applied to countries that have a different socioeconomic situation and that apply different policies may generate results with predictions that may be worse or better tailored depending on the context, an element that must be taken into consideration for the optimization of the modernization options to be used. In this sense, it is advisable to continue gathering information and contrasting the performance of the models in comparable situations to assess their predictive validity.

The usefulness and applicability of mathematical models lies in their ability not only to describe but also to forecast the evolution of the epidemic under alternative scenarios, with obvious and positive consequences in the control of the pandemic. Furthermore, if the predictions are consistent with data external to the model, then the hypotheses and parameter assumptions on which the model is based can be considered to be validated. In possible future scenarios like the current COVID-19 pandemic, predictive and explanatory models may be extremely useful to recommend prompt and optimal strategies in the early stages of outbreaks. The models can help provide a reduction in the spread and severity of these diseases in the context of very early and optimal implementation of health, social, and economic initiatives, particularly when initiatives are internationally coordinated to cover epidemics a of global nature.

## 5. Conclusions

Numerous methods for predicting infectious diseases can be found in the scientific literature, many of which, although known for decades, have been applied and improved during the current COVID-19 pandemic. Many of the first wave forecasts often provided a sound basis for action, although prediction of subsequent waves has remained somewhat elusive. There is no clear consensus on which prediction method is best, and it is assumed that one must tailor the approach to the context in addition to the underlying objectives being prioritized. The main difficulty of the comparison was that each method was tested in different countries, each from different sources of information, and with different characteristics and strategies for combating the disease. It is desirable to continue to compare the performance of the models in comparable situations, assessing their predictive validity. Based on the above, it will probably be feasible to make reliable and practical recommendations on their use in future outbreaks and pandemics.

## Figures and Tables

**Figure 1 ijerph-19-05546-f001:**
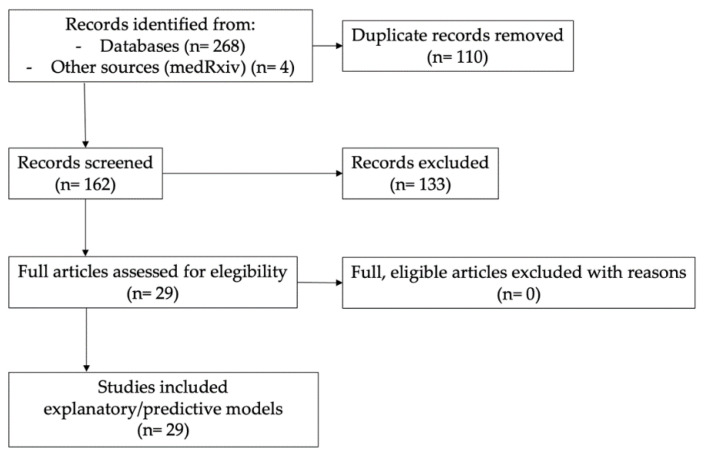
Flow diagram for the literature search and study selection.

**Figure 2 ijerph-19-05546-f002:**
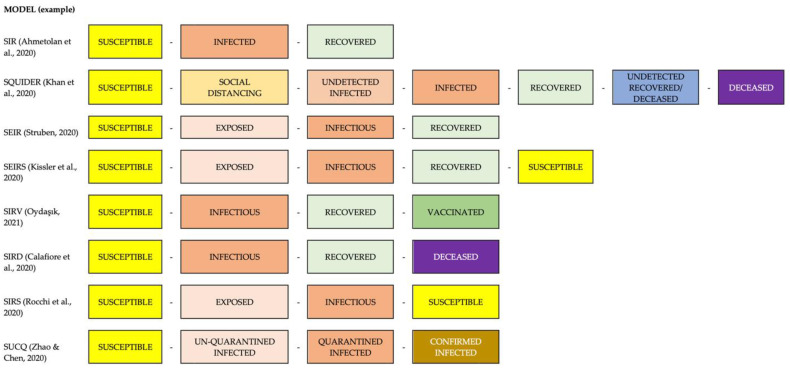
Parameters included in some of the main standard epidemiological models applied in COVID-19 prediction studies [18,19,20,21,22,23,24,25].

**Table 1 ijerph-19-05546-t001:** Main objectives and conclusions of relevant epidemiological models applied during the COVID-19 pandemic.

Reference	Model	Subjects	Objective	Time-Period	Results and Conclusions
**Fanelli & Piazza.**	SIR	COVID-19, China, Italy, France	Analyze the temporal dynamics of the coronavirus disease 2019 outbreak in China, Italy and France in the time	22 January 2020 to 12 March 2020	The kinetic the kinetic parameter that describes the rate of recovery seems to be the same, irrespective of the country, while the infection and death rates appear to be more variable. A simulation of the effects of drastic containment measures on the outbreak in Italy indicates that a reduction of the infection rate indeed causes a quench of the epidemic peak.
**Ahmetolan et al.**	SIR	Daily case reports and daily fatalities for China, South Korea, France, Germany, Italy, Spain, Iran, Turkey, the United Kingdom and the United States	The quantity that can be most robustly estimated from normalized data is shown to be the times of peak and the times of inflection points of the proportion of people infected. These values correspond to the peak of the epidemic and to the highest rates of increase and highest rates of decline in the number of people infected. The stability of the estimates is tested by comparing predictions based on data over long time periods.	January to May 2020	It is observed that the basic reproduction number and the mean duration of the infectious period can be estimated only in cases where the spread of the epidemic is over (for China and South Korea in the present case). Nevertheless, it is shown that the timing of the maximum and timings of the inflection points of the proportion of infected individuals can be robustly estimated from the normalized data. The validation of the estimates by comparing the predictions with actual data has shown that the predictions were realized for all countries except the USA, as long as lockdown measures were retained.
**Khan** et al.	SQUIDER	Detected and undetected infected populations, social sequestration, release from sequestration, plus reinfection; eight US states that make up 43% of the US population (Arizona, California, Florida, Illinois, Louisiana, New Jersey, New York State and Texas)	A compartmental model is proposed to predict the coronavirus 2019 (COVID-19) spread	22 January to 29 June 2020	Projections based on the current situation indicate that COVID-19 will become endemic. f lockdowns had been kept in place, the number of deaths would most likely have been significantly lower in states that opened up. Additionally, we predict that decreasing the contact rate by 10%, or increasing testing by approximately 15%, or doubling lockdown compliance (from the current ~15% to ~30%) will eradicate infections in Texas within a year. Extending our fits for all of the US states, we predict about 11 million total infections (including undetected), and 8 million cumulative confirmed cases by 1 November 2020. This model predicts significantly more COVID-19 cases and deaths, with an extended duration past two years for the majority of states examined.
**Cooper et al.**	SIR	Investigate the time evolution of different populations and monitor diverse significant parameters for the spread of the disease in various communities, represented by China, South Korea, India, Australia, USA, Italy and the state of Texas in the USA.	The effectiveness of the modelling approach on the pandemic due to the spreading of the novel COVID-19 disease. The authors propose predictions on various parameters related to the spread of COVID-19 and on the number of susceptible, infected and removed populations until September 2020	January to June 2020	If comparing the recorded data with the data from our modelling approaches, we deduce that the spread of COVID-19 can be under control in all communities considered, if proper restrictions and strong policies are implemented to control the infection rates early from the spread of the disease.
**Hauser et al.**	SEIR	Fitted transmission model to surveillance data from Hubei Province, China, and applied the same model to six regions in Europe: Austria, Bavaria (Germany), Baden-Württemberg (Germany), Lombardy (Italy), Spain, and Switzerland.	(1) Simulate the transmission dynamics of SARS-CoV-2 using publicly available surveillance data and (2) infer estimates of SARS-CoV-2 mortality adjusted for biases and examine the CFR, the symptomatic case-fatality ratio (sCFR), and the infection-fatality ratio (IFR) in different geographic locations.	January to May 2020	A comprehensive solution is proposed for the estimation of SARS-CoV-2 mortality from surveillance data during outbreaks. Asymptomatic case fatality rate (CFR) is not a good predictor of overall SARS-CoV-2 mortality and should not be used for policy evaluation or comparison between settings. Geographic differences in the infection-case fatality rate (IFR) suggest that a single IFR should not be applied to all settings to estimate the total size of the SARS-CoV-2 epidemic in different countries. The sCFR and IFR, adjusted for right-censoring and preferential determination of severe cases, are measures that can be used to improve and monitor clinical and public health strategies to reduce deaths from SARS-CoV-2 infection.
**Struben J.**	SEIR	South Korea, Germany, Italy, France, Sweden, and the United States	Develop a behavioral dynamic epidemic model for multifaceted policy analysis comprising endogenous virus transmission (from severe or mild/asymptomatic cases), social contacts, and case testing and reporting.	December 2019–15 May 2020	It determines how the timing and efforts of expanding testing capacity and reducing social contact interact to affect outbreak dynamics and can explain much of the cross-country variation in outbreak pathways. Second, in the absence of scaled availability of pharmaceutical solutions, post-peak social contacts should remain well below pre-pandemic values. Third, proactive (targeted) interventions, when supplemented by general deconfinement preparedness, can significantly increase eligible post-peak social contacts.
**Chen et al.**	Time-dependent SIR	China and extended to Japan, Singapore, South Korea, Italy, and Iran.	They propose a susceptible-infected-recovered (SIR) model that is time-dependent according to two time series: (i) transmission rate at time t and (ii) recovery rate at time t: (i) the transmission rate at time t and (ii) the recovery rate at time t. This approach is not only more adaptive than traditional static SIR models, but also more robust than direct estimation methods. Note: From data provided by the Health Commission of the People’s Republic of China (NHC).	12 February 2020	This time-dependent SIR model is not only more adaptive than traditional static SIR models, but also more robust than direct estimation methods. The numerical results show that one-day prediction errors for the number of infected persons X(t) and the number of recovered persons R(t) are within (almost) 3% for the dataset collected from the National Health Commission of the People’s Republic of China (NHC) [1]. Moreover, we are capable of tracking the characteristics of the transmission rate and the recovering rate with respect to time t, and precisely predict the future trend of the COVID-19 outbreak in China. To address the impact of asymptomatic infections in COVID-19, we extended our SIR model by considering two types of infected persons: detectable infected persons and undetectable infected persons. Whether there is an outbreak in such a model is characterized by the spectral radius of a 2 × 2 matrix that is closely related to the basic reproduction number R0.
**Calafiore** et al.	SIRD	Italy	Analyze parameters such as the initial number of susceptible people and the proportionality factor α (number of positives detected versus unknown number of infected people) to predict the spread of COVID-19	23 February to 30 March 2020	It was not possible to accurately calculate the variability of the results because of time restrictions, but it was estimated at ±78% based on previous sources
**Venkatesen M et al.**	SIR	India	The objective of this study is to provide a simple but effective explanatory model for the prediction of the future development of infection and for checking the effectiveness of containment and lockdown. A SIR model with a flattening curve and herd immunity based on a susceptible population that grows over time and difference in mortality and birth rates.	29 January to 15 April 2020	It illustrates how a disease behaves over time, taking variables such as the number of sensitive individuals in the community and the number of those who are immune. It accurately models the disease, considering the importance of immunization and herd immunity. The outcomes obtained from the simulation of the COVID-19 outbreak in India make it possible to formulate projections and forecasts for the future epidemic progress circumstance in India.
**Kuniya T.**	SIRS	Japan	Objective to give a prediction of the epidemic peak of COVID-19 in Japan using the real-time data, and taking into account the uncertainty due to the incomplete identification of the infected population.	1 January to 29 February 2020	R0 = 2.6 (95% CI, 2.4–2.8) is estimated, with an epidemic peak in the summer of 2020. Epidemiological conclusions: (1) the size of the essential epidemic is less likely to be affected by the rate of identification of the actual infectious population; (2) the intervention has a positive effect in delaying the peak of the epidemic; (3) intervention is needed over a relatively long period to effectively reduce the final size of the epidemic.
**Rocchi et al.**	SIRS	Italy	Objective: predict a potential scenario in which a balance is reached between susceptible, infected and recovered groups (something that usually occurs in epidemics).	15 April 2020	This model offers an analytical solution to the problem of finding possible steady states, providing the following equilibrium values: susceptible, about 17%, recovered (including deceased and healed) ranging from 79 to 81%, and infected ranging from 2 to 4%. However, it is crucial to consider that the results concerning the recovered, which at first glance are particularly impressive, include the huge proportion of asymptomatic subjects.

## Data Availability

Not applicable.

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
