# Peer review of "Predictive Models for Forecasting Public Health Scenarios: Practical Experiences Applied during the First Wave of the COVID-19 Pandemic"

_ijerph, 2022, doi:10.3390/ijerph19095546_

Round 1
Reviewer 1 Report
I was invited to revise the paper entitle "Predictive Models for Forecasting Public Health Scenarios: Practical Experiences Applied During the COVID-19 Pandemic". It was a systematic review that aimed to summarize main modeling approaches used during the first COVID-19 wave.
The topic is very interesting and faced off an important topic for public health.
Observations:
- Why Authors searched in medRvix and BiorXiv? These databases published preprint papers that need peer-review process;
- In PRISMA diagram Authors should report the exact number of paper identified for each sources;
- Results section is poor. Authors should not simply report the flow diagram but should describe better the inclusion process (Section 3.0);
- Table reported as Appendix B is very important and should be reported in the main text;
- Discussion section is too poor. Authors should better discuss about differences among models and focus on strenght and limitations of each ones;
- Authors should also discuss about differences among countries. Different policies can influence the spread of the pandemic and clearly influence models results. The same model cannot be applied in all countries that have different socio-economic status and that implemented different policies;
- Finally, Authors should highlight the usefullness of these results and how they can implemented in future pandemic.
Author Response
Dear Reviewer 1,
Thank you for your kindness and constructive feedback on our paper Manuscript ID: ijerph-1667277, entitled: “Predictive Models for Forecasting Public Health Scenarios: Practical Experiences Applied During the COVID-19 Pandemic.”
We have carefully considered and addressed each of your concerns.
Please see attached a detailed response to each of the comments / a point-by-point response to your remarks. We have tried to be as clear and as specific as possible in our response.
We are also sending two copies of the manuscript.
- One of is the revised manuscript as a clean copy.
- The other file related to the revised manuscript highlights the main changes introduced, presented in "track changes" mode.
We thank you, because we believe that your comments have helped us to revise and hopefully improve the manuscript.
We are looking forward to this paper´s potential publication in the IJERPH, and we will do our best to respond to any additional requests if deemed required.
Best wishes, also on behalf of all the co-authors,
Jose M Martin-Moreno, MD, PhD, DrPH

Reviewer 2 Report
Please see the attached file.

Author Response
Dear Reviewer 2,
Thanks for your thoughtful feedback on our paper Manuscript ID: ijerph-1667277, entitled: “Predictive Models for Forecasting Public Health Scenarios: Practical Experiences Applied During the COVID-19 Pandemic.”
We have done our utmost to address each of your concerns.
Please see attached a detailed response to each of the comments / a point-by-point response to your remarks. We have tried to be as clear and as specific as possible in our response.
We are also sending two copies of the manuscript.
- One of is the revised manuscript as a clean copy.
- The other file related to the revised manuscript highlights the main changes introduced, presented in "track changes" mode.
Thanks again for your comments, which have helped us to carefully revise and hopefully improve the manuscript.
We are looking forward to this paper´s potential publication in the IJERPH, and we will do our best to respond to any additional requests if deemed required.
Best regards, also on behalf of all the co-authors,
Jose M Martin-Moreno, MD, PhD, DrPH

Round 2
Reviewer 1 Report
The paper is now acceptable for publication. I want to congratulate with Authors for the excellent work.
Author Response
We would like here to simply thank you for your kind comments on our responses to the first round of suggestions, and the positive acceptance of our previous review.
Reviewer 2 Report
See the attached file

Author Response
Thank you again for your comments and suggestions. We have tried to respond appropriately and improve the original manuscript.
In the attached document, we respond point by point to your comments.
